# Loess Landslide Detection Using Object Detection Algorithms in Northwest China

Yuanzhen Ju [1], Qiang Xu [1,*], Shichao Jin [2], Weile Li [1], Yanjun Su [3,4], Xiujun Dong [1] and Qinghua Guo [5,6]

1   State Key Laboratory of Geohazard Prevention and Geoenvironment Protection, Chengdu University of Technology, Chengdu 610059, China; juyuanzhen@cdut.edu.cn (Y.J.); liweile08@mail.cdut.edu.cn (W.L.); dongxiujun@cdut.cn (X.D.)
2   Plant Phenomics Research Centre, Academy for Advanced Interdisciplinary Studies, Collaborative Innovation Centre for Modern Crop Production Co-Sponsored by Province and Ministry, Nanjing Agricultural University, Nanjing 210095, China; jschaon@njau.edu.cn
3   College of Resources and Environment, University of Chinese Academy of Sciences, Beijing 100049, China; ysu@ibcas.ac.cn
4   State Key Laboratory of Vegetation and Environmental Change, Institute of Botany, Chinese Academy of Sciences, Beijing 100093, China
5   Institute of Remote Sensing and Geographical Information Systems, School of Earth and Space Sciences, Peking University, Beijing 100871, China; guo.qinghua@pku.edu.cn
6   Institute of Ecology, College of Urban and Environmental Science, Peking University, Beijing 100871, China
*   Correspondence: xq@cdut.edu.cn

**Abstract:** Regional landslide identification is important for the risk management of landslide hazards. The traditional methods of regional landslide identification were mainly conducted by a human being. In previous studies, automatic landslide recognition mainly focused on new landslides distinct from the environment induced by rainfall or earthquake, using the image classification method and semantic segmentation method of deep learning. However, there is a lack of research on the automatic recognition of old loess landslides, which are difficult to distinguish from the environment. Therefore, this study uses the object detection method of deep learning to identify old loess landslides with Google Earth images. At first, a database of loess historical landslide samples was established for deep learning based on Google Earth images. A total of 6111 landslides were interpreted in three landslide areas in Gansu Province, China. Second, three object detection algorithms including the one-stage algorithm RetinaNet and YOLO v3 and the two-stage algorithm Mask R-CNN, were chosen for automatic landslide identification. Mask R-CNN achieved the greatest accuracy, with an AP of 18.9% and F1-score of 55.31%. Among the three landslide areas, the order of identification accuracy from high to low was Site 1, Site 2, and Site 3, with the F1-scores of 62.05%, 61.04% and 50.88%, respectively, which were positively related to their recognition difficulty. The research results proved that the object detection method can be employed for the automatic identification of loess landslides based on Google Earth images.

**Keywords:** loess landslide; google earth image; deep learning; automatic identification; object detection

## 1. Introduction

China has a vast territory, variable terrain, and frequent geological hazards that seriously threaten people's lives, especially landslides. For landslide hazard prevention, landslide identification is the basis for other research works. The identification of landslides can be divided into two categories. One is the identification of an indication before the landslide. Methods in this category are to observe the deformation of the slope based on multitemporal data. The object of the observation is the unstable slope that has not yet become a disaster [1], and the research methods include InSAR technology [2] and multi-period terrain data [3]. The other category is the identification of landslides that have already occurred. The results of landslide detection can be used for the study of

the susceptibility of landslides with other data, such as topography, landform, strata and lithology [4].

Optical remote sensing images have been widely applied in landslide identification, mapping, and monitoring. The methods of identifying historical landslides from remote sensing images can be generally divided into four groups: visual interpretation, pixel-based feature threshold, machine learning (ML), and deep learning (DL) methods. Visual interpretation outlines the landslide boundary based on the image texture differences between the landslide area and surrounding non-landslide area [5,6]. Visual interpretation relies heavily on the knowledge and experience of landslide professionals. The visual interpretation results are generally accurate. The pixel-based feature threshold method uses one or more thresholds in terms of different pixel information, such as the spectrum, texture, landform, and topography, to determine whether a pixel is a landslide or not [7–9]. The feature thresholds are usually determined according to regional landslide statistics, which may cause extrapolation issues to other regions. Machine learning and deep learning are two promising types of data-driven methods for landslide identification. This method is similar to the feature threshold and change detection methods during its data preparation. Machine learning methods classify landslides using various features relevant to landslides' occurrence [10–14]. The commonly used machine learning algorithms for landslide detection include Bayesian, logistic regression, support vector machine, random forest, and artificial neural networks. Deep learning is another powerful data-driven method of landslide detection. Compared with machine learning, deep learning methods with more hidden layers do not need to construct and select feature layers manually. The sample size in deep learning can be very large, making it more suitable for landslide identification across large regions. Convolution neural network (CNN) is the main method in landslide identification [15–20].

In the existing landslide identification research using deep learning, the research objects have been new landslides triggered by seismic landslides [15,20] and rainfall [19], which can be easily distinguished from the environment. There are few studies on old loess landslides, which are similar to the environment and have inconspicuous boundaries due to strong natural erosion and human transformation. The optical image data used in landslide identification research mainly include satellite images [15,17,18], aerial images [19], and unmanned aerial vehicle (UAV) images [15]. The terrain data are predominantly airborne lidar data [18]. For example, Landsat and Sentinel-2 images and STRM terrain data are available for free, but their spatial resolution is so low that it is difficult to detect small loess landslides and the fine features of landslides, e.g., the scarps and the cracks. Moreover, when old landslides have been seriously remodeled, their boundary features are also easily omitted. Hence, the use of low-spatial-resolution images for identification causes many omissions. Google Earth images are mosaic from pictures taken by a variety of satellite sensors or airplanes and can be obtained for free. Although the temporal and spatial resolutions of different regions are different, most regions still have a high-spatial-resolution, which can enable the identification of historical loess landslides. In image identification, deep learning has four main options: image classification, object detection, semantic segmentation, and instance segmentation [21]. Image classification aims to output a category label based on the input image. Object detection aims to automatically identify the category and location of the target object in the input image, and the location is usually the bounding rectangle where the target is located. Semantic segmentation tries to automatically identify the category and location of the target object in the input picture. The location is usually the irregular boundary of the target, but there is no distinguishable boundary between different objects of the same category. Instance segmentation is the integration of object detection and semantic segmentation, which can automatically identify the category and irregular boundary of each target object in the input image. In the research of landslide automatic identification using deep learning, image classification and semantic segmentation are usually included. In terms of image classification, Wang et al. [22] used CNN to identify new landslides in Hong Kong according to image classification,

and the final identification accuracy reached 92.5%. Ji et al. [17] used the attention boosted CNN model to recognize the data of new landslides in Guizhou Province according to image classification, and the final identification accuracy reached 96.62%. In terms of semantic segmentation, Yi and Zhang [23] used LandsNet to investigate the automatic identification of landslides triggered by earthquake, and the final F1-score was 86.89%. Qi et al. [24] used ResU-Net to analyze rainfall landslides in Tianshui City, Gansu Province, China, and the final F1-score was 89%. In the automatic identification of landslides, the disadvantage of image classification is that it is unable to determine the landslide location accurately. The semantic segmentation task obtains the landslide pixels. It is challenging to obtain the complete boundary of a single landslide in areas where landslides are dense and intersect with each other. Under these circumstances, the object detection and instance segmentation tasks can better realize the location and identification of a single landslide. There are some challenges to detecting landslides: (1) the lack of a database of old loess landslides with open source and high-spatial-resolution images for deep learning; and (2) a lack of evaluation of object detection algorithms applied in the loess landslide identification field.

Because of the above research gaps, this study aimed to recognize the old loess landslides automatically with Google Earth image data by the object detection algorithm of deep learning. Three study sites were selected in the Loess Plateau of China. The landslides at Site 1 are largely ancient landslides, some of which have been transformed by human activities. Site 2 is dominated by ancient seismic landslides, most of which have been transformed by human activities. Site 3 chiefly consists of landslides caused by loess erosion. Although the landslides in this area occurred later, they have been subjected to more natural transformations. Thus, there is little difference between them and the environment. Based on Google Earth images, remote sensing interpretation of historical loess landslides in these areas was carried out. A landslide identification sample database was established, and it was divided into a training set, validation set, and testing set. Then, a variety of object detection algorithms were trained and tested based on the sample library, including Mask R-CNN [25], RetinaNet [26], and YOLO v3 [27], and the identification accuracy of each algorithm was compared. Finally, the identification accuracy at different study sites was compared.

## 2. Data and Methods

### 2.1. Study Sites

This study selected three sites in northwestern China to study the detection of loess landslides (Figure 1). The three study sites are located in the Loess Plateau of China, characterized where landslides can be visually interpreted on optical remote sensing images because of the sparse vegetation. Peng et al. [28] divided the Loess Plateau of China into eight areas with a high incidence of loess landslides. Site 1 in this work is located in Areas I and II, Site 2 is located in Area IV, and Site 3 is located in Area V [28]. Site 1 was significantly influenced by the uplift of the Qinghai–Tibet Plateau and the earthquakes that occurred in Tianshui and Tongwei [29]. Many landslides at Site 1 were induced by earthquakes, and their landslide deposits have been utilized by human beings for building villages or farmlands while their landslide scarps were still identifiable. There are also some new loess landslides at Site 1 (Figure 2a). Most loess landslides at Site 2 were caused by the Haiyuan earthquake in 1920 [30,31], and their landslide bodies have also been severely modified by human activities. Since there has been a relatively short period for landslide evolution after the Haiyuan earthquake, the boundary features of the landslide source and accumulation areas at Site 2 are still preserved to a certain extent (Figure 2b). Most loess landslides at Site 3 were caused by gravitational erosion, resulting in steep scarps and distinct landslide accumulation areas (Figure 2c). The spectral characteristics of loess landslides are generally similar to those of the nearby environment at the three study sites.

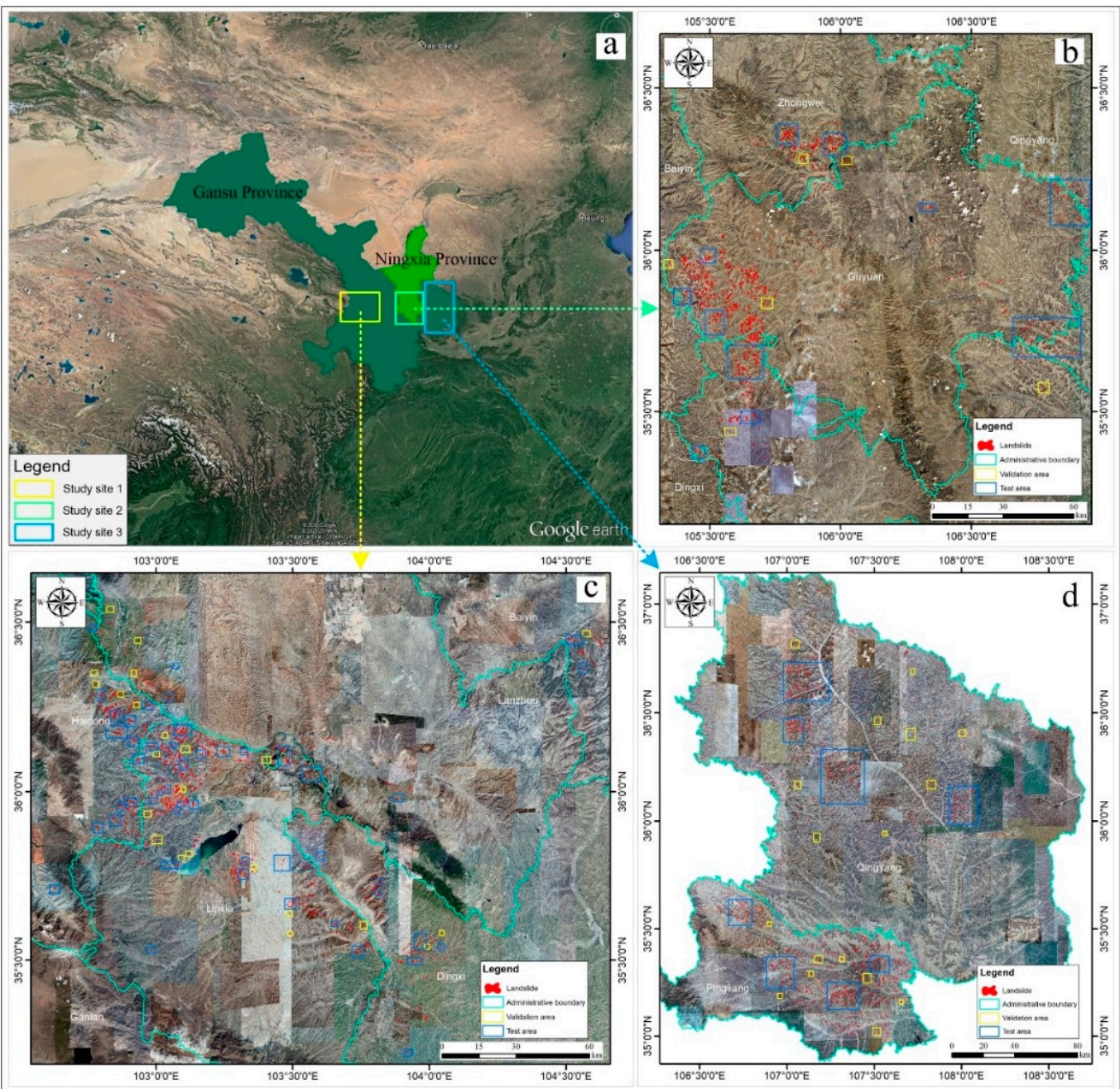

**Figure 1.** Location of study sites: (**a**) location of study site on Google Earth; (**b**) the distribution of landslides at study site 2; (**c**) the distribution of landslides at study site 1; (**d**) the distribution of landslides at study site 3.

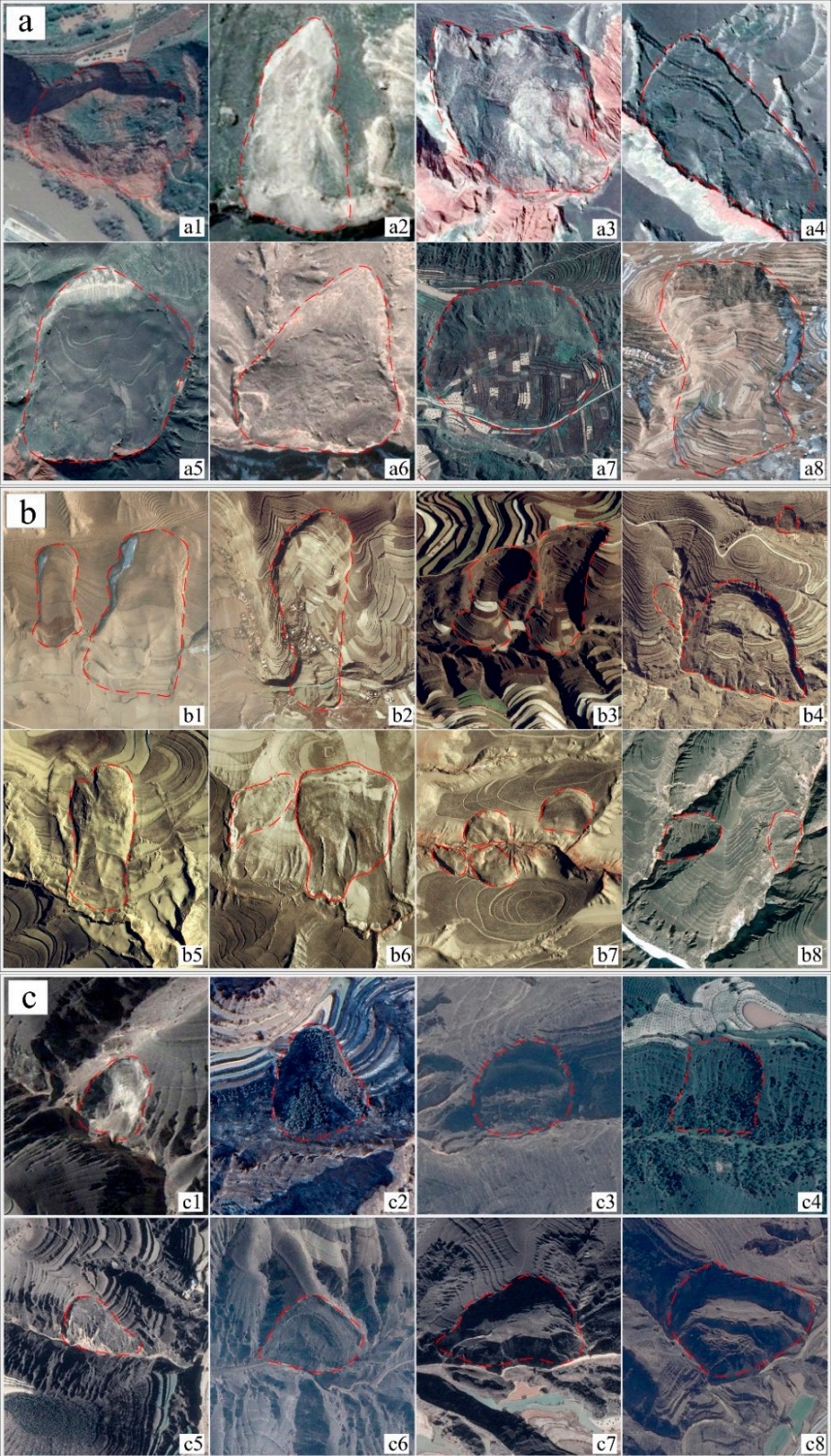

**Figure 2.** Samples of loess landslides. (**a**) Study site 1, a1—a3: the different forms of the new landslides; a4: the creep landslide; a5—a6: the unchanged ancient landslides; a7—a8: the ancient landslides. (**b**) Study site 2, b1—b8: the co-seismic landslides changed by human activity induced by Haiyuan earthquake. (**c**) Study site 3, c1—c8: the landslides caused by gravity erosion and slope erosion.

## 2.2. Data Labeling and Data Set Division

The optical remote sensing images used in this study were collected from Google Earth. Since Google Earth mosaics images from various satellite sensors, the spectral characteristics, spatial resolutions, and dates of images used in this work are unavoidably inconsistent. The spatial resolution of most images in this study was 1 m × 1 m, and the image dates ranged from 2015 to 2018. A visual interpretation was used to identify and draw landslides in this study with the aid of ArcMap. To ensure landslide labeling quality, three landslide experts conducted cross-validation of the landslide interpretation results. In total, this study labeled 6111 landslides, including 2683 landslides at Site 1, 1452 landslides at Site 2, and 1976 landslides at Site 3. The landslide areas varied from 1290 $m^2$ to 829,380 $m^2$. The longest and shortest sides of the rectangular boundaries of the landslides were 1484 m and 47 m, respectively (Table 1).

**Table 1.** Datasets of Landslide Samples.

| Study Area | Dataset | | | Landslide Area | | | Bounding Box Length | | |
|---|---|---|---|---|---|---|---|---|---|
| | Training | Validation | Testing | Max Area ($10^3$ $m^2$) | Min Area ($10^3$ $m^2$) | Average Area ($10^3$ $m^2$) | Max Length (m) | Min Length (m) | Average Length (m) |
| Site 1 | 1875 | 61 | 747 | 829.38 | 1.29 | 51.97 | 1388 | 88 | 340 |
| Site 2 | 870 | 39 | 543 | 607.37 | 1.53 | 63.18 | 1484 | 76 | 400 |
| Site 3 | 1118 | 59 | 799 | 749.41 | 4.11 | 71.55 | 1130 | 47 | 266 |
| Total | 3863 | 159 | 2089 | 829.38 | 1.29 | 51.97 | 1484 | 47 | 400 |

In this study, loess landslide detection was formatted as an object detection task. The original polygons of landslide labels were transformed into the COCO labeling format required by the object detection algorithm (i.e., the outer rectangle of the landslide boundary). Then, the landslide data set was randomly divided into training, validation, and testing data sets. There were 3863 training samples, 159 validation samples, and 2089 testing samples (Table 1). The training data set was utilized for training the deep learning model. The validation data set was employed to select the optimal model. The testing data set was used to evaluate the performance of the optimal model. Considering the variety of landslide shapes and sizes and the hue difference in input image data, multiple verification areas and test areas (Figure 1) were chosen at the study sites to evaluate the method more comprehensively and objectively. To detect the landslide and non-landslide areas at the study sites, the Google Earth images of validation and test areas were cropped into the patches of 2000 pixel × 2000 pixel from the upper left corner. To prevent the landslide samples in the validation and testing areas being cut into two pieces, the overlap size in the cutting direction was set to 500 pixels based on the average length of landslides.

## 2.3. Deep Learning Methods

Three image-based object detection DL models were used for loess landslide detection in this study, namely Mask R-CNN [25], RetinaNet [26], and YOLO v3 [27]. These algorithms belong to anchor-based methods [32], which specify initial object boundary boxes of different sizes and proportions according to prior experience. Mask R-CNN is a two-stage algorithm, which employs the region proposal network (RPN) [33] to filter the initial object boundary boxes and then perform classification, coordinate regression, and mask segmentation on the filtered boundary boxes. RetinaNet and YOLO v3 belong to one-stage algorithms. After obtaining the initial object boundary boxes, the boundary boxes are directly classified, and the boundary regression is performed to obtain the category and accurate position of the object. RetinaNet uses focal loss to deal with the imbalance of positive and negative samples in training.

The training and testing of the DL models were performed using a desktop with main hardware parameters as follows: Intel Xeon e5-2640 CPU, 256 GB RAM, and NVIDIA Tesla P100-PCIE-12GB GPU. The code of the three models originated from Github (https://github.com/open-mmlab/mmdetection, accessed on 10 February 2022). It was built based on the framework of Pytorch. ResNet-101 and FPN were selected as the backbone

network in Mask R-CNN (Figure 3). The minimum batch quantity was set at 2 images. The training batch was 100, and each batch was iterated 1000 times. The learning rate was 0.005, with a learning momentum of 0.9. The weight regularization coefficient was 0.005. During training, all the samples were resized to 1024 × 1024. The RetinaNet model used ResNet-101 and FPN as the backbone network (Figure 3). The learning rate was 0.00001. The minimum number of batches was 8. The gamma parameter of the focal loss was 2.0, and the alpha parameter was 0.25. All the samples were resized to 1024 × 1024 during training. The YOLO v3 used DarkNet-53 as the backbone network (Figure 3). The minimum batch quantity was 4 and the learning rate was 0.001, with a learning momentum of 0.9. All the samples were resized to 1024 × 1024 during training.

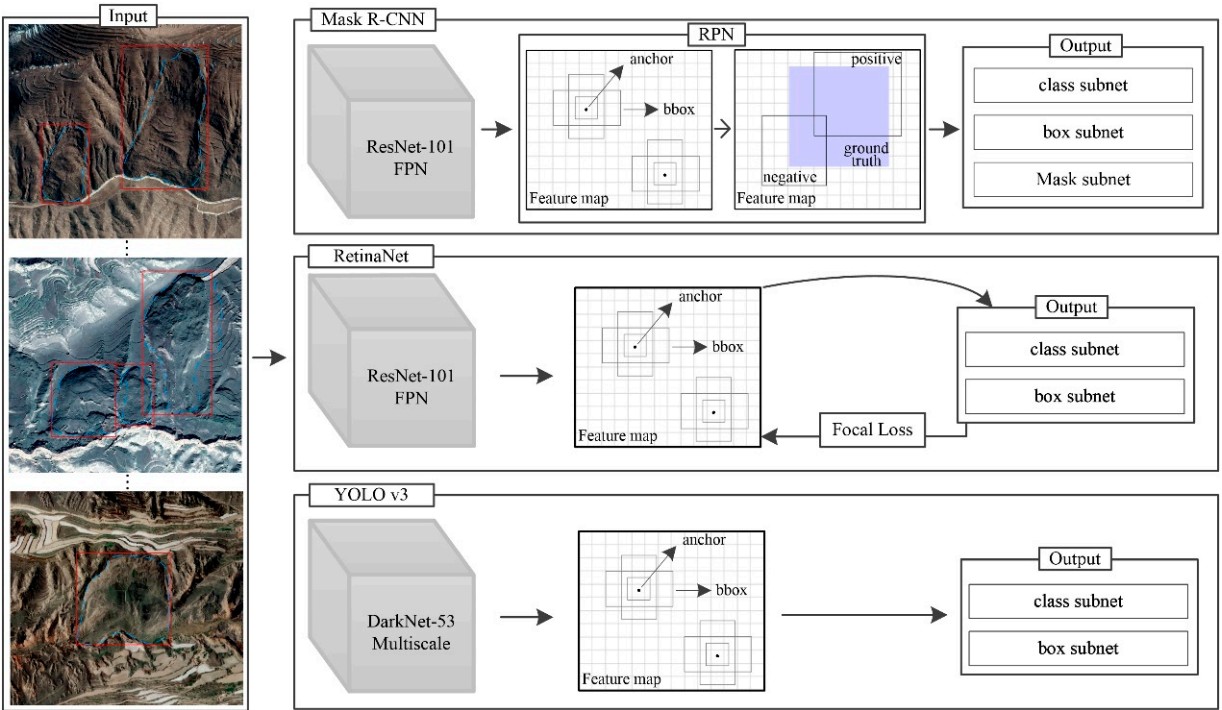

**Figure 3.** Flowchart of object detection model used for landslide detection.

To increase the generalization ability of the DL models and reduce the training time, the initial parameters of the DL models were obtained by the COCO pre-training model. The validation data set was used to optimize the model parameters of the DL models. The model with the greatest accuracy on the validation data set was selected as the optimal model, whose performance was evaluated using the held-out testing data set.

*2.4. Accuracy Evaluation*

Average precision (AP) and mean average precision (mAP) are widely used to evaluate the accuracy of object detection projects. AP is calculated using the recall–precision curve of a certain category under a given IoU (intersection of union), which is the ratio of the intersection area to the union area of the prediction and the ground truth. The area under the recall–precision curve is the AP value. The mAP is the average value of the average precision among multiple classes. There was only one object class for our landslide detection problem; thus, AP was utilized as the performance metric in this work. Specifically, AP50 and AP (0.05:0.5:0.95) were selected. AP50 was the AP value when the IoU threshold was set to 0.5. AP (0.05:0.5:0.95) was the average AP value when the step size was 0.05 and the IoU threshold was increased from 0.5 to 0.95. AP (0.05:0.5:0.95) is usually directly recorded as AP. We used AP and AP50 to choose the best model on validation dataset, and compared the performance of different models.

In addition, we used F1-score to evaluate different landslide detection methods. The F1-score is a comprehensive index that considers both precision and recall, and is computed as follows:

$$F1 - score = \frac{2 * P * R}{P + R} \qquad (1)$$

where $P$ is the precision and $R$ is the recall. Precision is the ratio of true positive samples to the sum of true positive and false positive samples. Recall is the ratio of true positive samples to the sum of true positive and false negative samples. For our two-class (landslide or non-landslide) classification task, $P$ was the ratio of the number of correct predictions to the number of all predictions, and $R$ was the ratio of the number of correct predictions to all ground truths.

We calculated the F1-score under a given confidence score and IoU. Many landslides in this work have a large head scarp and multiple minor scarps. As shown in Figure 4, the most easily identified position is the scarp, and the accumulation area with an ambiguous boundary is missing. So, we chose a lower IoU threshold to calculate the F1-score, which was 0.3, compared to the index of AP. We tried different confidence score thresholds to calculate the F1-score accuracy, and then chose the threshold corresponding to the best accuracy on the validation dataset. The confidence score threshold of Mask R-CNN was 0.3, of RetinaNet was 0.15, and of YOLO v3 was 0.05.

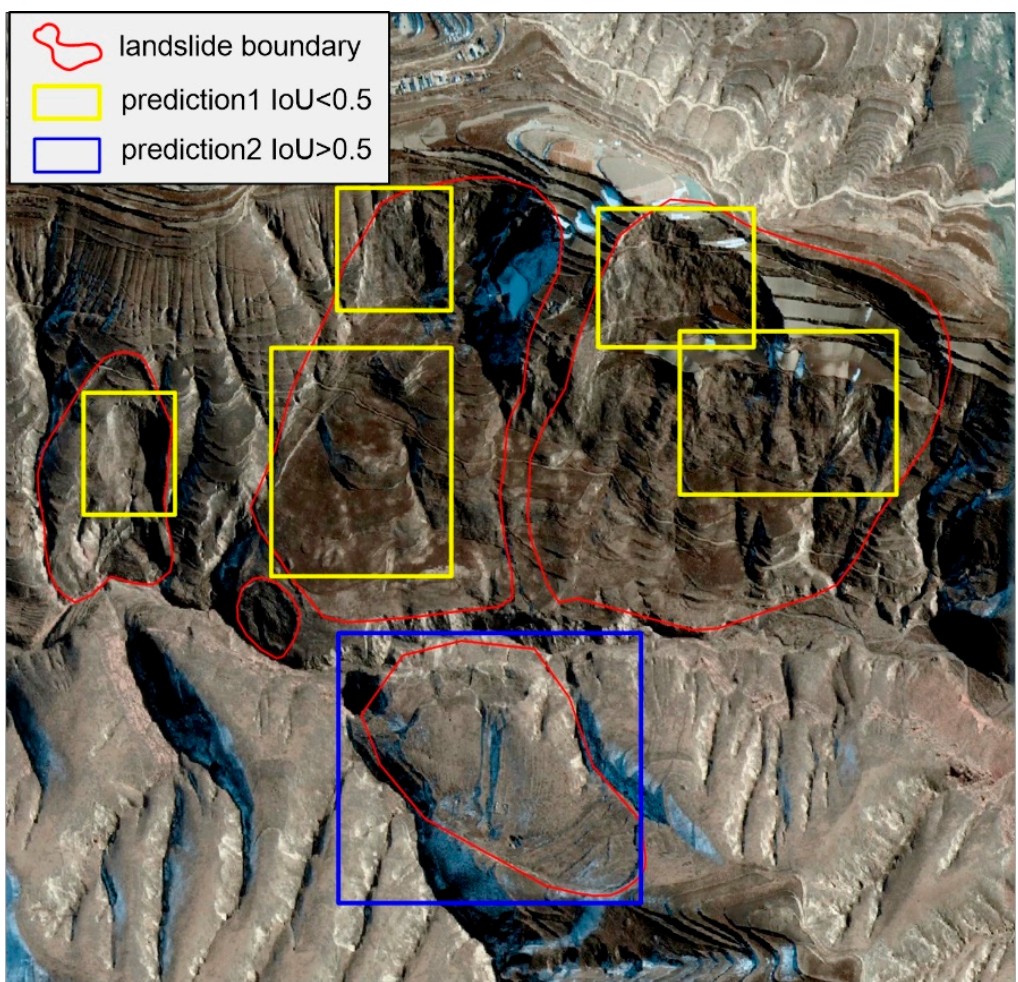

**Figure 4.** Results of a multi-stage landslide.

As described in Section 2.2, we cut the remote sensing images of the validation and test area into smaller patches of 2000 pixels × 2000 pixels with an overlap size of

500 pixels × 500 pixels. We calculated the AP on the original results of the patches and F1-score on the fusion results.

## 3. Results

In the training process, the model with the best AP accuracy evaluated on the validation set was chosen for the test. The test accuracies of the three DL models (i.e., Mask R-CNN, RetinaNet, and YOLO v3) are shown in Table 2. Generally, the two-stage model of Mask R-CNN outperformed the RetinaNet and YOLO v3 models. The two-stage model of Mask R-CNN yielded AP and AP50 values of 18.9% and 35.7%, respectively. The one-stage models of YOLO v3 and RetinaNet showed AP values of 15.5% and 17% and AP50 values of 31.5% and 32.3%, respectively. When F1-scores were used to evaluate the accuracy of the fusion result on Mask R-CNN, the F1-score was 55.31%, and the corresponding precision and recall were 47.41% and 66.37%, respectively. The F1-score of the RetinaNet model was 47.53%, and the corresponding precision and recall were 45.8% and 49.4%, respectively. The F1-score of the YOLO v3 model was 46.99%, and the corresponding precision and recall were 42.63% and 52.34%, respectively.

**Table 2.** Accuracy of the Test Dataset.

| Model | COCO Evaluation | | Score Threshold | F1-Score Evaluation | | |
|---|---|---|---|---|---|---|
| | AP | AP50 | | Precision | Recall | F1-Score |
| Mask R-CNN | 18.9% | 35.7% | 0.3 | 47.41% | 66.37% | 55.31% |
| RetinaNet | 17.0% | 32.3% | 0.15 | 45.80% | 48.40% | 47.07% |
| YOLO v3 | 15.5% | 31.5% | 0.05 | 43.63% | 55.34% | 48.79% |

Figure 5 shows the landslide detection results of the three DL models against the test data set. The red polygons represent the ground-truth boundary of landslides, while the yellow rectangles represent the predicted landslide boundaries. Figure 6 suggests that Mask R-CNN had the fewest prediction results, while RetinaNet and YOLO v3 gave more prediction frames. Since Google Earth images are composed of images from different sensors at different dates, there is much hue difference.

The optimal DL model, the Mask R-CNN model, was used to compare its detection abilities across different environments. The precision, recall, and F1-score of the Mask R-CNN model at the three study sites are shown in Table 3. The results suggest the Mask R-CNN model achieved higher accuracy at Sites 1 and 2 than at Site 3. The hightest F1-score was 62.05% at Site 1, with the precision of 57.42% and the recall of 67.5%. The lowest F1-score was 50.88%, with the precision of 41.35% and the recall of 66.11%. Example landslide detection results of the Mask R-CNN model at the three study sites are shown in Figure 6.

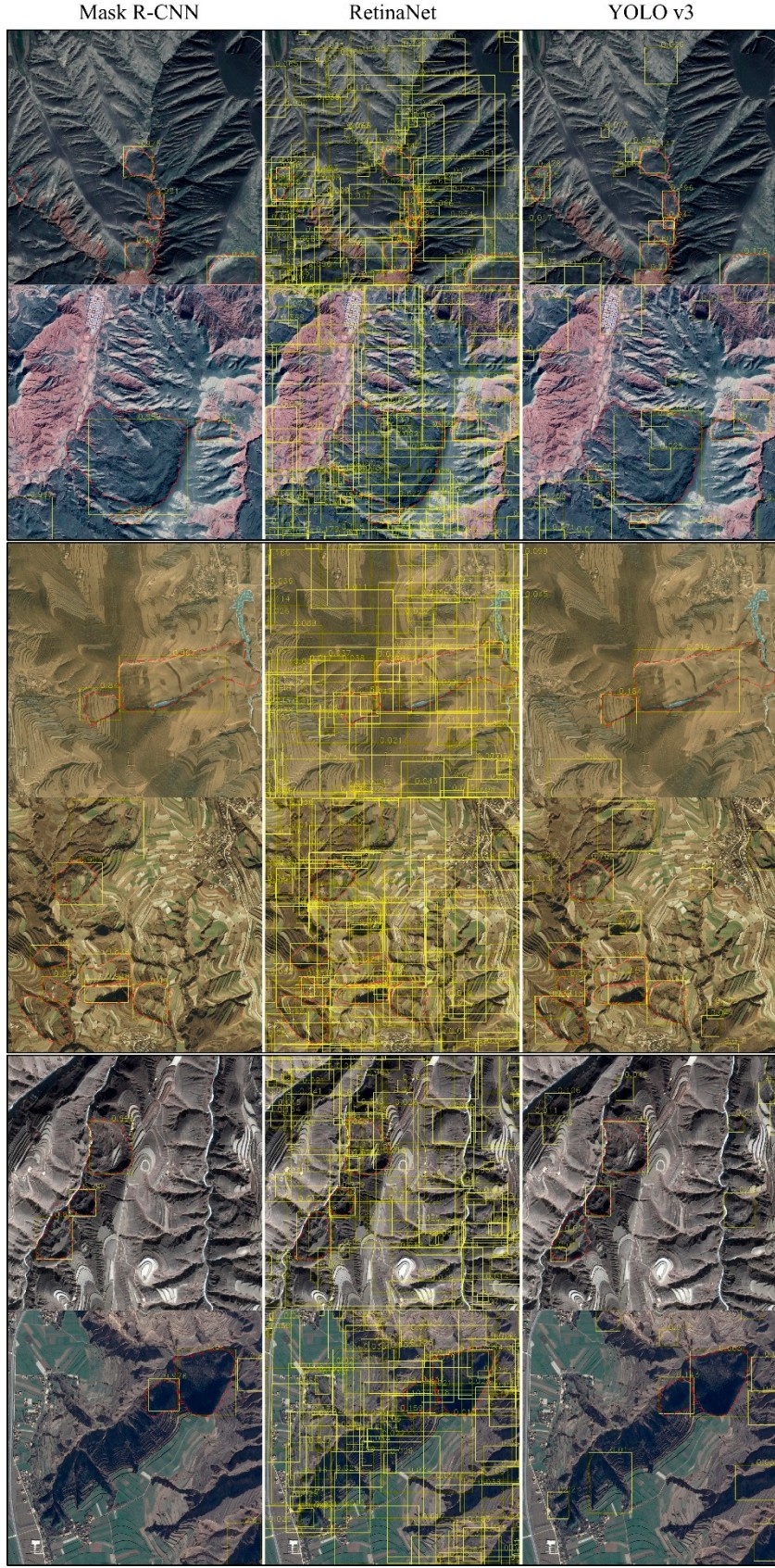

**Figure 5.** Results of different models.

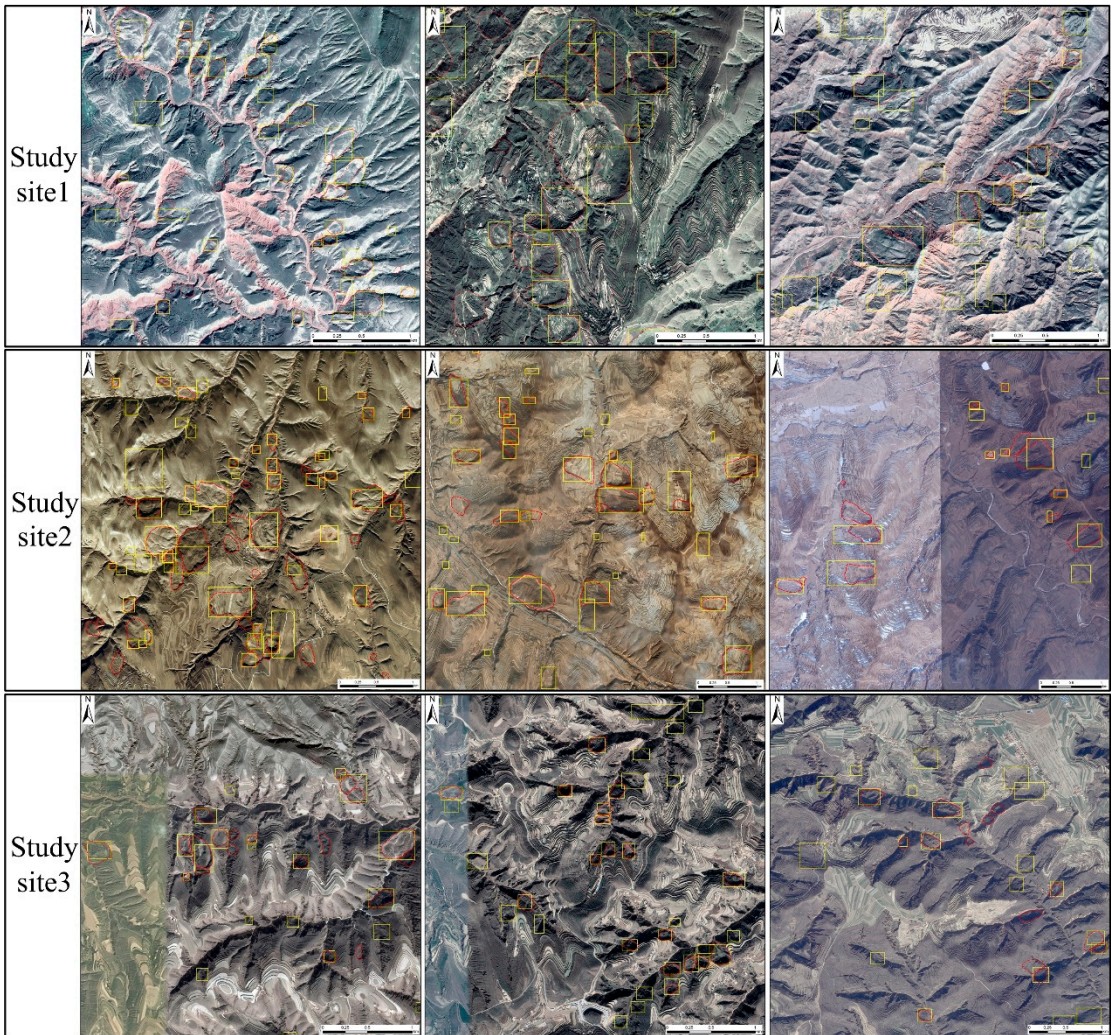

**Figure 6.** Landslide detection results of the mask R-CNN model against different study areas.

**Table 3.** Recall, precision, and F1-score of Mask R-CNN model against different study sites.

| Study Area | Precision | Recall | F1-Score |
|------------|-----------|--------|----------|
| Site 1 | 57.42% | 67.50% | 62.05% |
| Site 2 | 52.13% | 73.62% | 61.04% |
| Site 3 | 41.35% | 66.11% | 50.88% |

## 4. Discussion

### 4.1. Landslide Detection Accuracy of Different Models

This study used three DL models to detect loess landslides. The main difference between the three models is that YOLO v3 and RetinaNet belong to a one-stage algorithm, while Mask R-CNN belongs to a two-stage algorithm. The accuracy of the two-stage algorithm was found to be better than that of the one-stage algorithm, which is consistent with other object detection tasks, such as face detection.

Figure 7 is the violin plot of the landslide area of detection and omission in the test dataset for the three models. The omission number of Mask R-CNN was the least, and RetinaNet was close to YOLO v3, which was identical to the trend in recall. In the detection of large landslides, Mask R-CNN has the strongest ability, followed by RetinaNet and YOLO v3. Mask R-CNN uses the feature cropped from feature maps by RoI (region of interest) to predict the position and category of an object in the head net. The RoI of a big object is more accurate than that of a small one, which makes a better performance when

detecting big objects in Mask R-CNN. RetinaNet densely predicts the target on the entire image, directly using the output features of the backbone. So, the accuracy of RetinaNet was lower than that of Mask R-CNN. YOLO v3 uses the last three feature layers of the backbone, and RetinaNet uses the last four. The neck net of RetinaNet can combine more fine semantic features than YOLO v3 can. So, RetinaNet had a better performance than YOLO v3.

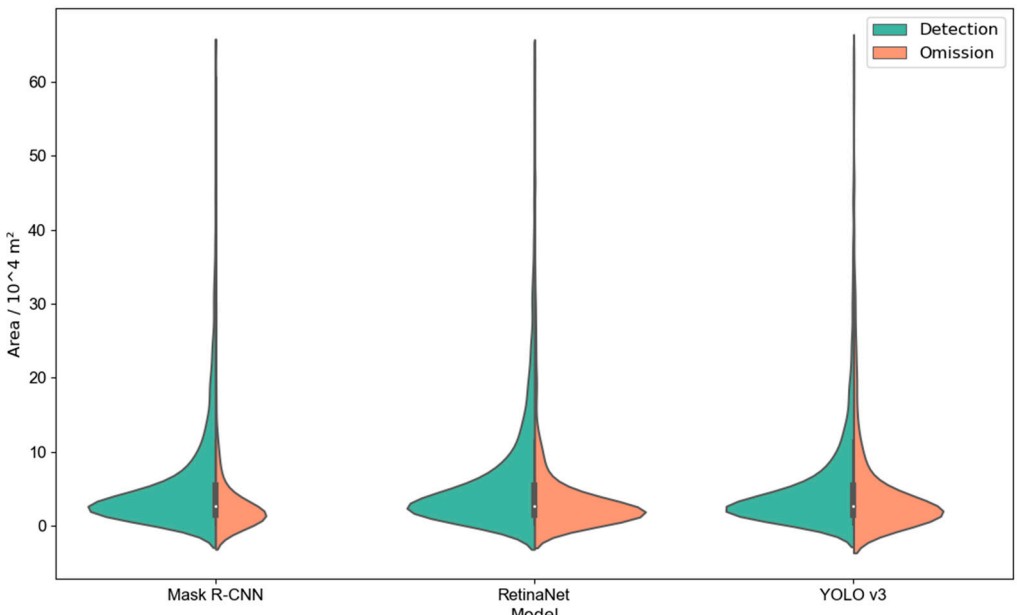

**Figure 7.** The violin plots of the landslide area of detection and omission in the test dataset for the three models.

Old loess landslides with Google Earth image achieved a lower detection accuracy compared to the new landslide detection [34]. Considering the data source, the spatial resolution of the Google Earth images was uneven, and the spatial resolution of some areas was lower. From the perspective of the identification target, the main target of the study was the old landslides that had been affected by human activities and/or natural erosion. The spectrum differences between loess landslides and their background are smaller when compared with those of the new landslides, which makes the automatic identification of loess landslides in the study areas a challenging task.

### 4.2. Identification Accuracy of Different Study Sites

The deep learning models generally exhibited good abilities in identifying the image textures of the arc-shaped or armchair-like scarps of landslides, but also tended to make mistakes in classifying non-landslide areas with similar texture features as landslides. Figure 8 shows example images of false positive predictions. In Figure 8a, the valley was identified as a landslide, possibly because the top and the ridge of the mountain formed an armchair-like boundary, and the valley showed different spectral characteristics when compared with the nearby environment. The false detections in Figure 8b mainly occurred in the shadowed parts of the image (mostly consisting of ridges or steep slopes), which may have been affected by the image brightness. The false identifications in Figure 8c mainly occurred on the terraces. The terraces and shadowed or eroded gullies could have formed image features similar to those of the old landslides, leading to false predictions. In Figure 8d, the DL model falsely identified slope sections where severe soil erosion occurred.

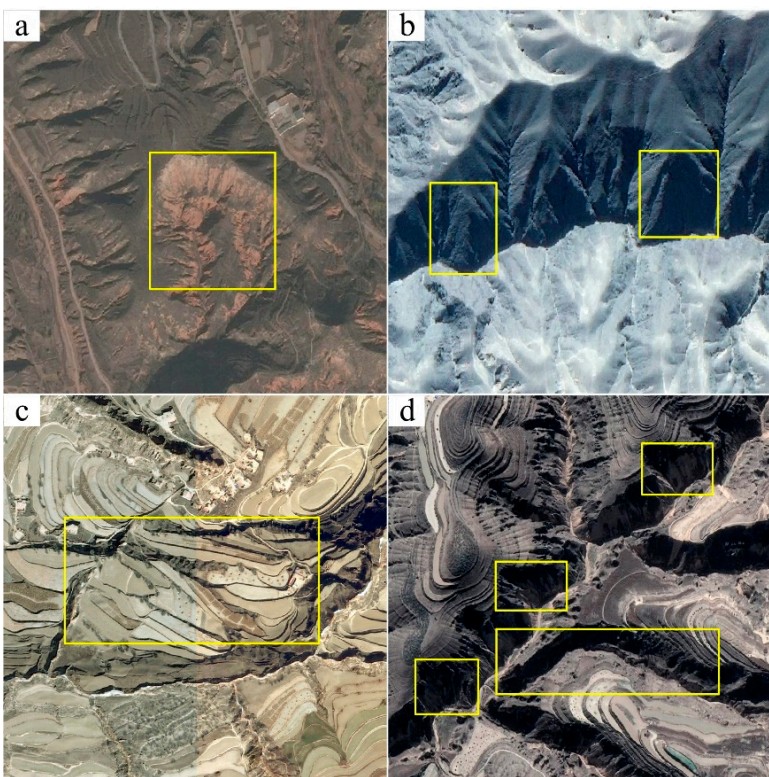

**Figure 8.** Example images showing false positive predictions: (**a**) in the valley; (**b**) in the shadow section; (**c**) in the terraces; (**d**) in the soil erosion section.

The optimal DL model showed the highest F1-score accuracy at Site 1, followed by Site 2 and Site 3. Figure 6 demonstrates that there were more types of landslides at Site 1, including ancient landslides and some new landslides. Among them, the scarp of the reactivated landslide was evident and was more distinguishable from the environment. Site 2 was mostly composed of coseismic landslides caused by the Haiyuan earthquake that occurred about 100 years ago. Most of these landslides were modified into farmland and villages. Because these landslides were relatively young, their landslide boundaries were still clear. Loess erosion phenomena were common at Site 3. The armchair morphology caused by soil erosion was similar to the morphology of landslides, which made it challenging to distinguish non-landslide slope erosions from loess landslides (Figure 8c,d).

### 4.3. Multi-Category and Multi-Position Landslide Detection Based on Multi-Sourced Data

According to the above research results, it was effective to utilize deep learning methods to identify old loess landslides. The historical landslides in different scenes could be identified to some extent, but the samples, data sources, and methods still need to be improved.

From the perspective of sample labeling, the landslide samples in this study were not classified in more detail. The next step could be to label landslides by their landslide types based on multiple information. First, the time of the landslide and the existence of human modification could be labeled. Landslides could be divided into new landslides, old landslides whose surface morphology has not been modified, and old landslides whose surface morphology has been modified. In this way, the automatic identification results could provide a better foundation for the study of landslides. Second, landslide elements, such as cracks, scarps, and accumulation areas, could be labeled. Figure 4b depicts that, compared to the complex landslide as a whole, the features of the steep scarp at the landslide source area could more easily be detected. Furthermore, the identification of cracks at the source area of an old landslide could also provide evidence of the revival of the old landslide. Finally, multiple types of non-landslide images that easily cause misidentification

could be constructed to provide more specific negative sample information for the model. In this work, the automatic identification results of historical loess landslides were classified into two categories: landslide and non-landslide. In the future, a multi-class landslide automatic detection model could be built based on the above multi-class label scheme. The multi-class task could help the detection model to learn more specific features relevant to different types of landslides.

From the perspective of data sources, multi-sourced data could be integrated into automatic landslide detection. Although optical remote sensing images from Google Earth are easy to obtain and their overall spatial resolutions are high, they have limited band information and are affected by cloud coverage. Other remote sensing images could be used to ensure image quality and DL-based landslide detection. For example, Sentinel 2 image data could be considered to enhance the diversity of the data spectrum and improve the accuracy of landslide identification. In addition to optical images, other three-dimensional data (e.g., 3D points data from LiDAR survey) could be integrated in the future. Furthermore, the DL models need to be trained to detect some key features of landslides, such as the main scarps of landslides.

## 5. Conclusions

In this study, a sample set of historical loess landslides was first constructed for deep learning. The samples were from three study sites, mainly consisting of old landslides with long sliding times. Open source Google Earth images were used as the data source, and three deep learning methods of object detection, namely, Mask R-CNN, RetinaNet, and YOLO v3, were utilized to automatically identify the ancient loess landslides. Mask R-CNN achieved the best performance, with an AP of 18.9% and F1-score of 55.31%. It showed that the two-stage object detection method is more suitable to detect old loess landslide. The above results indicate that, based on the deep learning method, Google Earth images can be used to automatically identify most of the historical loess landslides at the study sites. The identification of landslides is the basis for regional landslide risk management. The early identification of landslides can also be realized in combination with other monitoring technologies, because a lot of landslides are clustered and reborn. Especially in the era of big data, the use of deep learning methods for landslide identification could substantially enhance the detection efficiency, which is of great significance to the prevention and mitigation of landslides.

**Author Contributions:** Y.J. conceived the manuscript; Y.J. collected the image data and interpreted the landslides; Q.X. provided funding support and ideas; S.J. helped to train the models; X.D., W.L., Q.G. and Y.S. helped to improve the manuscript. All authors have read and agreed to the published version of the manuscript.

**Funding:** The work was funded by the National Innovation Research Group Science Fund of China (Grant No. 41521002), the National Key Research and Development Program of China (Grant No. 2021YFC3000401), the Sichuan Science and Technology Support Plan (No. 2018SZ0339) and The National Natural Science Foundation of China (No. 42072306).

**Institutional Review Board Statement:** Not applicable.

**Informed Consent Statement:** Not applicable.

**Data Availability Statement:** The optical remote sensing images used in this study were provided by Google Earth.

**Acknowledgments:** We are very grateful to optical image data provided by Google Earth. We thank the reviewers and the editor for their constructive comments and suggestions, which would significantly improve this paper, and thank Yao Guangle, Zhan Weiwei and Zhou Wenlong for their help. We also would like to express our gratitude to EditSprings for the expert linguistic services provided.

**Conflicts of Interest:** The authors declare no conflict of interest.

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
