# Peer review of "Loess Landslide Detection Using Object Detection Algorithms in Northwest China"

_remotesensing, doi:10.3390/rs14051182_

Round 1

Reviewer 1 Report

The authors apply 3 different state-of-the-art object detectors to images obtained from Google Earth for ‘loess landslides’ detection.

I’m sorry to say that the manuscript is quite unreadable and then incomprehensible in several of its parts, it is mainly because:

1) terms like scale, statistic, spectral, and many more, are used too freely, without following a standard or accepted ontology.

2) the construction of many sentences is very contorted and confusing (probably the linguistic service was only meant to fix simple grammar errors)

The two points make it difficult, if not impossible, to evaluate the contents of this work. My perception is about a set of boxes not well or properly connected: remotely sensed data, geomorphology, and machine (deep) learning, and the result is a purely numeric exercise. The authors, in my opinion, should spend time understanding the problems in the data and how they propagate in the final results and connect the different ingredients when possible.

My last general comment is about the chosen special issue. The authors should try and contextualize this work in the framework of engineering geology remote sensing, by showing how the working scale can be of help for engineering applications (probably very hard to do).

In the end, in my opinion, the work does not seem to have any scientific soundness and it is not properly written, so my suggestion is to reject it. I encourage the authors to reformulate the scientific framework by better amalgamating the different components and to invest time in the writing style.

Here are my comments in detail. Note that I stopped to go into details of the writing starting from par. 3

Introduction

34 threaten the safety of people’s lives and property: threaten people’s lives.

34-35 especially landslide hazards that frequently occur: especially landslides, unless you want to explicit what a landslide hazard that frequently occurs is.

35-36 landslide identification or landslide mapping is the basis for…: it sounds like identification and mapping are interpreted as if they were the same thing, but they are not.

36 – 38 According to the development stage of landslides, the identification of landslides can be divided into two categories: one is the early identification of landslides and the other is the identification of historical landslides: a sentence difficult to interpret, I suggest to declare what some terms mean, in an accepted ontological context. What is a development stage of a landslide? What is ‘early identification’ in this context? Does it mean identification just after the occurrence of the slope failure or identification of precursors? Are all the landslides already occurred ‘historical landslides’?

39 – 40 (1) Study of the susceptibility of landslides: I think this is improper, susceptibility is not identification, but spatial forecasting, usually for shallow landslides.

41 – 42 It is the earliest and is chiefly based on the identification of historical landslides: again, not sure about this, see Reichenbach et al 2018. Many other inventories are used, including multitemporal.

42-43 Observation of the dynamic changes of slopes based on multitime series data: can changes by static? What is ‘multitime’?

45 – 46 Identification of the overall or local revival of historical landslides: what does it mean? Are you talking about local or catchment scale landslide activity?

47 – 48 This is a comprehensive judgment in the same location through the static identification of historical landslides and the dynamic observation of new landslides: what does it really mean?

48 – 50 Therefore, the identification of historical landslides is the basis for the early identification of landslides: the connection between the two identifications is actually not explicit, and it cannot be deducted from the previous sentences.

51 – 52 emphasizes the time resolution of data: what does it mean?

52 – 53 The identification of historical landslides can be completed by single period optical image data: single period??

55 large-scale geological hazards: or geological hazards affecting large areas?

58 – 79: a lot of confusion. To put a bit of order, I suggest having a look at some review papers on methods for landslide detection/mapping. The choice is wide.

81 – 83 Compared with machine learning, deep learning does not need to construct and select feature layers manually when landslide features are processed: this is actually not necessarily true for several nets, including U-net (but not only).

83 accepted: other ML can accept the same quantity of data, I guess here you mean needed.

88 – 90 In the existing landslide identification research using deep learning, the research objects are primarily seismic landslides (Ghorbanzadeh et al., 2019a; Yu et al., 2020) and rainfall landslides (Sameen and Pradhan, 2019): another misleading sentence: many landslides are triggered by rainfall or earthquakes, including, probably those landslides studied here.

90 – 91 textural characteristics are quite different from the environment: please, elaborate on the concept.

92 which were occurred for a long time: please, re-phrase

99 resolution is low: for identification, this is a relative concept, it depends on the type and size of the landslide. I suggest linking this sentence to the next one

107 tasks: options?

119 identify the data of new landslides: what does it mean?

124 semantic landslides: what does it mean?

129 -130 precise boundary of a single landslide belonging to the same type: what does it mean?

133 -134 lack of the application of object detection algorithms in landslide identification field: not sure I understand what this means.

2. Data and Methods

2.1 study sites

154 The landslide samples were distributed in three study sites: distributed?

155 – 156 the average annual rainfall is small, the evaporation is large, the climate is dry, and the vegetation is lacking: small, large… compared to what?

156 therefore: what is the connection between the two sentences? Optical images can be used in several contexts.

158 – 159 The scale of loess landslides varies significantly, ranging from several kilometers to tens of meters: the scale or the size??

163 affected: disturbed? Modified?

167 – 168 landslides induced by the erosion of rainfall, groundwater, and river: I’m not an expert in geomorphology, but I’m quite sure this sentence does not make much sense.

175 The landslides caused by gravity erosion: what?

175 obvious: clear?

2.2. Data Labeling and Data Set Division

190 The image data of landslide interpretation in this study were Google Earth images: of?

192 and time of large-scale images may be inconsistent: scale? Pixel size?

194 – 196 The marked content was the fine polygon vector boundary of landslides. After the landslide marking was completed, three landslide experts would cross-check and determine the final remote

sensing interpretation results: marked content … would … remote sensing interpretation: too poorly written.

207 variety of landslide types: early all classified as loess landslides.

207 inconsistent spectrum of image data: inconsistent? Do you mean that images are not radiometric corrected?

210 According to the statistics …: this is not statistics… just the largest and the smallest sizes

214 statistical information: see my previous comment, furthermore, it is not clear the connection (according…) between the choice of the box size (4.000.000 m2) and the previous numbers. The largest landslide is 830.000 m2, so? Shall the patch be 5 times larger than the largest landslide? Why?

How about the small ones?

219 – 221 To avoid the landslide samples in the validation and

testing areas to be cut into two pieces, the overlap size in the cutting direction was set to

500 pixels: how did you choose 500? Trial process, or was it based on the minimum distance between two landslides?

2.3. Deep Learning Methods

249 – 250 All the samples were reclassified into 1024 × 1024 during training: why reclassified?? and why 1024, and how? I guess re-sampled.

256 To increase the generalization ability of the model: this concept should be better introduced, since loess landslides are not in COCO, and it does not seem to me that being able to detect the categories in COCO can be an added value. Please discuss this.

258 – 260 Meanwhile, to prevent the overfitting phenomenon, the validation set data were used to verify the accuracy of the model during training. The model with the greatest accuracy of

the validation set was considered as the optimal model and was used to test the data: maybe I don’t get it correctly, but to me this is wrong, this is not the way to prevent overfitting. Overfitting is minimised with l1, l2, or dropout.

2.4 Accuracy evaluation

General comment: the entire paragraph is to be restructured and simplified by stating clearly how to use AP for the best selection of the threshold values.

265 – 267 When evaluating the accuracy of object detection, we need to set the confidence score threshold and IoU (Intersection of Union) threshold to determine whether or not a certain prediction result is correct: quite confusing. Confidence score and IoU allow evaluating the accuracy of the model once they are fixed.

268 this: which one?

268 It is used to evaluate the reliability of the result: I don’t think ‘reliability’ is correct, the sentence itself is ambiguous. It just says how likely the anchor box contains an object.

278 – 279 The AP aims to calculate the recall and precision of a certain category under a given IoU threshold: it does aim to calculate recall and precision but it makes use of them.

288 – 289 The essence of the AP indicator is to consider all predictions that are available (the confidence score threshold is assumed to be 0.01): what does it mean?

288 – 295: I feel like there is a contradiction here: it was not suitable for, but then it is used… Actually here and the entire paragraph can be simplified. The output of the model is a probability, and the AP curves can help to find the best threshold of the confidence score, varying the IoU…

290 – 291 it was not suitable for the detection of old landslides here because the error results output by the model accounted for a large proportion: I can’t understand

298 – 300: I’m not sure this means Precision = tp/(tp+fp), and Recall = tp/(tp+fn)

301 latest: the most recent one?

Fig.4: it actually seems to me that an anchor box grabbing the big landslides is missing as if the contextualization advantages of these methods were not exploited properly.

I’ll stop here to check in detail the sentences

3 Results

313 of:?

319 According to actual needs: like?

319 – 321 when the F1 score, which is more suitable for old landslides (multistage landslide or multiple sliding landslide): what does it mean?

Fig 5: what does the 45deg straight line mean? This is not ROC

340 their spectral features are not uniform: not sure about the meaning, also because CNN's work more with spatial features.

341 – 342 in different spectral regions: visible? Infrared?

346 – 347 the F1-score index was more appropriate for practical application: what does it mean?

4 discussions

4.1. Identification Accuracy of Different Models

General comments:

1) It is not discussed why Yolo performs better than the others.

2) all the problems listed here (Resolution, dis-homogeneity of the images, weak traces left by landslides eroded, and so on) are quite typical problems in machine learning-based landslide detection, any connection to the models used here is missing. I would have expected some reasoning in relation to the size of the anchor boxes and the sizes of the FP and FN (just an example), or how these problems can affect those specific models.

380 – 382 To improve the accuracy in the application, we also chose F1-score to evaluate the accuracy of old landslide detection, which could filter the results that were lower than the confidence score threshold: F1 is a way to evaluate the performance, it does not improve the accuracy!!

4.2 Identification Accuracy of Different Study Sites

395 – 397 Due to the steep slope, it is

easy to form shadows similar to the scarp of the landslide under the sun, thus resulting in

false identifications: meaning that the Nets use shadows to detect landslides.

4.3. Multi-category and Multi-position Landslide Detection Based on Multi-sourced Data

427 can provide a better foundation for the study of landslide activity: I would not use ‘activity’ here.

430 – 432 Furthermore, the identification of the cracks at the landslide source area can also provide a basis for the early identification of the landslide revival: what does it mean?

438 – 440 It also can limit the previously encountered misclassification to some categories and can give a greater degree of trust to the easier categories: can →  may, actually this is a supposition.

I don’t think, at this point, being useful to review the abstract and conclusions because the paper does not have enough consistency for that.

Author Response

Dear Editors and Reviewers:

Thank you for your letter and for the reviewers’ comments concerning our manuscript entitled “Loess landslide detection using object detection algorithms in Northwest China” (ID: remotesensing-1531838). Those comments are all valuable and very helpful for revising and improving our paper, as well as the important guiding significance to our research. We have studied comments carefully and have made corrections which we hope meet with approval. Line numbers used in the reply section refer to the text in the revised manuscript.

The main corrections in the paper and the responds to the reviewer’s comments are as following:

Reviewer #1

General comments:

The authors apply 3 different state-of-the-art object detectors to images obtained from Google Earth for ‘loess landslides’ detection. I’m sorry to say that the manuscript is quite unreadable and then incomprehensible in several of its parts, it is mainly because:

1) terms like scale, statistic, spectral, and many more, are used too freely, without following a standard or accepted ontology.

2) the construction of many sentences is very contorted and confusing (probably the linguistic service was only meant to fix simple grammar errors)

3) The authors should try and contextualize this work in the framework of engineering geology remote sensing, by showing how the working scale can be of help for engineering applications (probably very hard to do).

Detail comments:

  1. Introduction

(1) 34 threaten the safety of people’s lives and property: threaten people’s lives.

Response: We have corrected it.

(2) 34-35 especially landslide hazards that frequently occur: especially landslides, unless you want to explicit what a landslide hazard that frequently occurs is.

Response: We have corrected it.

(3) 35-36 landslide identification or landslide mapping is the basis for…: it sounds like identification and mapping are interpreted as if they were the same thing, but they are not.

Response: We have corrected it.

(4)

36 – 38 According to the development stage of landslides, the identification of landslides can be divided into two categories: one is the early identification of landslides and the other is the identification of historical landslides: a sentence difficult to interpret, I suggest to declare what some terms mean, in an accepted ontological context. What is a development stage of a landslide? What is ‘early identification’ in this context? Does it mean identification just after the occurrence of the slope failure or identification of precursors? Are all the landslides already occurred ‘historical landslides’?

39 – 40 (1) Study of the susceptibility of landslides: I think this is improper, susceptibility is not identification, but spatial forecasting, usually for shallow landslides.

41 – 42 It is the earliest and is chiefly based on the identification of historical landslides: again, not sure about this, see Reichenbach et al 2018. Many other inventories are used, including multitemporal.

42-43 Observation of the dynamic changes of slopes based on multitime series data: can changes by static? What is ‘multitime’?

45 – 46 Identification of the overall or local revival of historical landslides: what does it mean? Are you talking about local or catchment scale landslide activity?

47 – 48 This is a comprehensive judgment in the same location through the static identification of historical landslides and the dynamic observation of new landslides: what does it really mean?

48 – 50 Therefore, the identification of historical landslides is the basis for the early identification of landslides: the connection between the two identifications is actually not explicit, and it cannot be deducted from the previous sentences.

51 – 52 emphasizes the time resolution of data: what does it mean?

Response: We have some mistakes about the concepts of early identification and susceptibility. We corrected and simplify this paragraph. Please see Line 37-46 in the revised manuscript

(5)

52 – 53 The identification of historical landslides can be completed by single period optical image data: single period??

55 large-scale geological hazards: or geological hazards affecting large areas?

58 – 79: a lot of confusion. To put a bit of order, I suggest having a look at some review papers on methods for landslide detection/mapping. The choice is wide.

83 accepted: other ML can accept the same quantity of data: I guess here you mean needed.

Response: We have reorganized this paragraph. Please see Line 47-73 in the revised manuscript

(6)81 – 83 Compared with machine learning, deep learning does not need to construct and select feature layers manually when landslide features are processed: this is actually not necessarily true for several nets, including U-net (but not only).

Response: Some algorithms with shallow neural network are built based on handcrafted features due to lack of effective image representation. Maybe it is up to the task complexity and the model representation. We have rewritten the sentence as “Compared with machine learning, deep learning methods with more hidden layers do not need to construct and select feature layers manually” Please see Line 68-69 in the revised manuscript

(7)88 – 90 In the existing landslide identification research using deep learning, the research objects are primarily seismic landslides (Ghorbanzadeh et al., 2019a; Yu et al., 2020) and rainfall landslides (Sameen and Pradhan, 2019): another misleading sentence: many landslides are triggered by rainfall or earthquakes, including, probably those landslides studied here.

Response: We have rewritten the sentence as “In the existing landslide identification research using deep learning, the research objects are the new landslides triggered by seismic landslides (Ghorbanzadeh et al., 2019a; Yu et al., 2020) and rainfall (Sameen and Pradhan, 2019)”. Please see Line 74-76 in the revised manuscript

(8)90 – 91 textural characteristics are quite different from the environment: please, elaborate on the concept.

Response: We have rewritten the sentence  as “they can be easily distinguished from the environment”. Please see Line 76-77 in the revised manuscript

(9)92 which were occurred for a long time: please, re-phrase

Response: We have rewritten the sentence  as “There are few studies on old loess landslides, which were similar to the environment with the inconspicuous boundary,”. Please see Line 77-78 in the revised manuscript

(10)99 resolution is low: for identification, this is a relative concept, it depends on the type and size of the landslide. I suggest linking this sentence to the next one

Response: We have rewritten the sentence  as “For example, Landsat and Sentinel-2 images and STRM terrain data are available for free, but their spatial resolution is so low that it is difficult to detect the small loess landslides and the fine features of landslide, eg. the scarps and the cracks.” Please see Line 83-86 in the revised manuscript.

(11)107 tasks: options?

Response:

(12)119 identify the data of new landslides: what does it mean?

Response: We have modified “identify the data of new landslides” to “identify new landslides”. Please see Line 106 in the revised manuscript.

13)124 semantic landslides: what does it mean?

Response: We have deleted the “semantic”.

(14)129 -130 precise boundary of a single landslide belonging to the same type: what does it mean?

Response: We have modified the sentence to “It is challenging to obtain the complete boundary of a single landslide in areas where landslides are dense and intersect with each other.” We have corrected it. Please see Line 116-117 in the revised manuscript.

(15)133 -134 lack of the application of object detection algorithms in landslide identification field: not sure I understand what this means.

Response: We have modified the sentence to “lack of the evalution of object detection algorithms applied in loess landslide identification field.” Please see Line 121-122 in the revised manuscript.

  1. Data and Methods

2.1 study sites

(1)

154 The landslide samples were distributed in three study sites: distributed?

155 – 156 the average annual rainfall is small, the evaporation is large, the climate is dry, and the vegetation is lacking: small, large… compared to what?

156 therefore: what is the connection between the two sentences? Optical images can be used in several contexts.

Response: changed “The landslide samples were distributed in three study sites (Fig. 1), all of which were located in the Loess Plateau of China. In these areas with, the characteristics of small average annual rainfall is small, the large evaporation is large, the dry climate is dry, and the lack of vegetation is lacking. Therefore, the remote sensing interpretation of loess landslides can be conducted arried out based on optical images” to “The landslide samples were distributed in three study sites (Fig. 1), all of which were located in the Loess Plateau of China with the characteristics of sparse vegetation due to less average annual rainfall, more evaporation and drier climate, compared to Southern China. Therefore, the optical images in RGB band can be used to conduct the visual interpretation of loess landslides in study area with less occlusion on ground”

(2)158 – 159 The scale of loess landslides varies significantly, ranging from several kilometers to tens of meters: the scale or the size??

Response: We have modified these sentences to “This study selected three sites in northwestern China to study detection of loess landslides (Fig. 1). The three study sites are located in the Loess Plateau of China characterized where landslides can be visually interpreted on optical remote sensing images because of the sparse vegetation..” Please see Line 142-145 in the revised manuscript.

(3)163 affected: disturbed? Modified?

Response: We have modified this sentence to “Site 1 was significantly influenced by the uplift of the Qinghai-Tibet Plateau and the earthquakes that occurred in Tianshui and Tongwei”. Please see Line 148-149 in the revised manuscript.

(4)167 – 168 landslides induced by the erosion of rainfall, groundwater, and river: I’m not an expert in geomorphology, but I’m quite sure this sentence does not make much sense.

Response: We have modified “There are also some new landslides induced by the erosion of rainfall, groundwater, and river” to “There are also some new landslides”. Please see Line 152-153 in the revised manuscript.

(5)175 The landslides caused by gravity erosion: what?

175 obvious: clear?

Response: We rewritten the paragraph. Please see line 142-161 in the revised manuscript.

2.2. Data Labeling and Data Set Division

(6)190 The image data of landslide interpretation in this study were Google Earth images: of?

Response: Changed “The image data of landslide interpretation in this study were are from Google Earth images” to “This study use Google Earth images to interpret landslides”

(7)192 and time of large-scale images may be inconsistent: scale? Pixel size?

Response: We rewritten the sentence. Please see line 175-177 in the revised manuscript.

changed “time of large-scale images” to “acquisition time of images in a wide range of area”

(8)194 – 196 The marked content was the fine polygon vector boundary of landslides. After the landslide marking was completed, three landslide experts would cross-check and determine the final remote sensing interpretation results: marked content … would … remote sensing interpretation: too poorly written.

Response: We have rewritten the sentence as “To ensure the landslide labeling quality, three landslide experts conducted cross-validation of the landslide interpretation results.” Please see line 178-179 in the revised manuscript.

(9)207 variety of landslide types: early all classified as loess landslides.

Response: We changed “landslide types” to “landslides shapes”.

(10)207 inconsistent spectrum of image data: inconsistent? Do you mean that images are not radiometric corrected?

Response: We changed “inconsistent spectrum” to “hue difference”

(11)210 According to the statistics …: this is not statistics… just the largest and the smallest sizes

Response: We removed the word “statistics”.

(12)214 statistical information: see my previous comment, furthermore, it is not clear the connection (according…) between the choice of the box size (4.000.000 m2) and the previous numbers. The largest landslide is 830.000 m2, so? Shall the patch be 5 times larger than the largest landslide? Why?How about the small ones?

Response: We choose the image size 2000 pixel × 2000 pixel based on the max length (1484 m) of landslides. So the net can get the field of a whole landslide. And it can get enough negative samples from the background. We changed “the remote sensing image was cut into 2000 pixel × 2000 pixel” to “the remote sensing image was cut into 2000 pixel × 2000 pixel based on the max length of landslides”.

(13)219 – 221 To avoid the landslide samples in the validation and testing areas to be cut into two pieces, the overlap size in the cutting direction was set to 500 pixels: how did you choose 500? Trial process, or was it based on the minimum distance between two landslides?

Response: We set 500 pixels based on the average length of landslides.

2.3. Deep Learning Methods

(14)249 – 250 All the samples were reclassified into 1024 × 1024 during training: why reclassified?? and why 1024, and how? I guess re-sampled.

Response: We changed “reclassified” to “resize”.

(15)256 To increase the generalization ability of the model: this concept should be better introduced, since loess landslides are not in COCO, and it does not seem to me that being able to detect the categories in COCO can be an added value. Please discuss this.

Response: We used COCO pre-training model to initialize the parameters when training a new model. The output layer parameters about categories of pre-training would be excluded. So We just used the backbone and head. There are about 200 thousand images in COCO dataset. The amount of COCO dataset is much larger than the loess landslide dataset in this study. So the pre-training model contains more information to extract the low and high dimension features.

(16)258 – 260 Meanwhile, to prevent the overfitting phenomenon, the validation set data were used to verify the accuracy of the model during training. The model with the greatest accuracy of the validation set was considered as the optimal model and was used to test the data: maybe I don’t get it correctly, but to me this is wrong, this is not the way to prevent overfitting. Overfitting is minimised with l1, l2, or dropout.

Response: The statement about overfitting phenomenon in the paper is wrong. The validation set was just used to pick the optimal model. We deleted “to prevent the overfitting phenomenon”

2.4 Accuracy evaluation

(17)

General comment: the entire paragraph is to be restructured and simplified by stating clearly how to use AP for the best selection of the threshold values.

265 – 267 When evaluating the accuracy of object detection, we need to set the confidence score threshold and IoU (Intersection of Union) threshold to determine whether or not a certain prediction result is correct: quite confusing. Confidence score and IoU allow evaluating the accuracy of the model once they are fixed.

268 this: which one?

268 It is used to evaluate the reliability of the result: I don’t think ‘reliability’ is correct, the sentence itself is ambiguous. It just says how likely the anchor box contains an object.

278 – 279 The AP aims to calculate the recall and precision of a certain category under a given IoU threshold: it does aim to calculate recall and precision but it makes use of them.

288 – 289 The essence of the AP indicator is to consider all predictions that are available (the confidence score threshold is assumed to be 0.01): what does it mean?

288 – 295: I feel like there is a contradiction here: it was not suitable for, but then it is used… Actually here and the entire paragraph can be simplified. The output of the model is a probability, and the AP curves can help to find the best threshold of the confidence score, varying the IoU…

290 – 291 it was not suitable for the detection of old landslides here because the error results output by the model accounted for a large proportion: I can’t understand

298 – 300: I’m not sure this means Precision = tp/(tp+fp), and Recall = tp/(tp+fn)

301 latest: the most recent one?

Fig.4: it actually seems to me that an anchor box grabbing the big landslides is missing as if the contextualization advantages of these methods were not exploited properly.

Response: AP and F1-score are two indicators to evaluation the model. AP is comprehensive and strict due to considering many IoU threshold with minimum of 0.5. F1-score is intuitive to observe the results of landslide detection. We have restructured the entire paragraph. Please see line 243-278 in the revised manuscript.

3 Results

(1)313 of:?

Response: changed the sentence “In the training process of the three models of Mask R-CNN, Retinanet, and YOLO v3” to “In the training process”

(2)319 According to actual needs: like?

Response: We delete the improper sentence.

(3)319 – 321 when the F1 score, which is more suitable for old landslides (multistage landslide or multiple sliding landslide): what does it mean?

Response: We delete the improper sentence.

Fig 5: what does the 45deg straight line mean? This is not ROC

Response: We found the P-R curve was inappropriate. We deleted the Fig 5.

(4)340 their spectral features are not uniform: not sure about the meaning, also because CNN's work more with spatial features.

Response: We have changed “their spectral features are not uniform” to “there are much hue difference”

(5)341 – 342 in different spectral regions: visible? Infrared?

Response: It is incorrect statement, We have removed it.

(6)346 – 347 the F1-score index was more appropriate for practical application: what does it mean?

Response: It is incorrect statement, We have removed it.

4 discussions

4.1. Identification Accuracy of Different Models

General comments:

(1)It is not discussed why Yolo performs better than the others.

Response: We checked the data and the design and parameters of the three model. The YOLO v3 stemmed from GitHub (https://github.com/ultralytics/yolov3) used more augmentation trips and the local loss. The code framework of the RetinaNet and YOLO v3 is pytorch, and Mask R-CNN is tesnsorflow. We retrained the three models from the same GitHub project (mmdetection), and removed all data augmentation trips. The Mask R-CNN achieved the best performance, followed by RetinaNet and YOLO v3. We updated all results.

(2) all the problems listed here (Resolution, dis-homogeneity of the images, weak traces left by landslides eroded, and so on) are quite typical problems in machine learning-based landslide detection, any connection to the models used here is missing. I would have expected some reasoning in relation to the size of the anchor boxes and the sizes of the FP and FN (just an example), or how these problems can affect those specific models.

Response: We have added the analysis on the landslide area of detection and omission in test sataset for the three models. Please see line 325-337 in the revised manuscript.

(3)380 – 382 To improve the accuracy in the application, we also chose F1-score to evaluate the accuracy of old landslide detection, which could filter the results that were lower than the confidence score threshold: F1 is a way to evaluate the performance, it does not improve the accuracy!!

Response: We agree with the reviewer. We have deleted the improper statement.

4.2 Identification Accuracy of Different Study Sites

(4)395 – 397 Due to the steep slope, it is easy to form shadows similar to the scarp of the landslide under the sun, thus resulting in false identifications: meaning that the Nets use shadows to detect landslides.

Response:  We have modified the paragraph. Please see Line 350-362 in the revised manuscript

4.3. Multi-category and Multi-position Landslide Detection Based on Multi-sourced Data

(5)427 can provide a better foundation for the study of landslide activity: I would not use ‘activity’ here.

Response: We have deleted the word “activity”.

(6)430 – 432 Furthermore, the identification of the cracks at the landslide source area can also provide a basis for the early identification of the landslide revival: what does it mean?

Response: We have changed the above sentence to “Furthermore, the identification of the cracks on the same position with the images obtained from different time can also provide evidence of landslide revival”. Please see Line 392-394 in the revised manuscript.

(7)438 – 440 It also can limit the previously encountered misclassification to some categories and can give a greater degree of trust to the easier categories: can →  may, actually this is a supposition.

Response: We have changed “can” to “may”.

Reviewer 2 Report

The paper is interesting and addresses a current issue of great significance in everyday life.
In this study, a set of historical loess landslide samples was first constructed for deep learning. The samples were from three study sites, consisting mainly of old landslides with a long slip time. The open source Google Earth image was used as a data source, and three deep learning methods for detecting objects, namely MASKR-CNN, RetinaNet and YOLO v3, were used to automatically identify old loess landslides.
The results presented by the authors indicated that, based on the deep learning method, Google Earth images can be used to automatically identify most historical loess landslides in the studied sites, but there are also problems of harmonizing images or rather their resolutions. , especially since images of different resolutions are used from different satellites and I recommend that this method be used on a wide scale, but in parallel with other established methods.
The identification of historical landslides is the basis of the regional management of landslide risk, the images of Google Earth being more recent and they must be corroborated with older data to obtain a multicriteria management and have a solid history.
Early identification of landslides can also be done in combination with other monitoring technologies, as many landslides are grouped and reborn and early identification can prevent many disasters that occur due to lack of real-time landslide monitoring.
Especially today and in the age of big data that can be updated and processed, using the deep learning method to identify landslides can substantially increase efficiency, which is of great importance for preventing and mitigating landslides.
I congratulate the authors of the article and it is a research that helps the management progress in monitoring landslides.
The article can be published in the journal.

Author Response

Dear Editors and Reviewers:

Thank you for your letter and for the reviewers’ comments concerning our manuscript entitled “Loess landslide detection using object detection algorithms in Northwest China” (ID: remotesensing-1531838). Those comments are all valuable and very helpful for revising and improving our paper, as well as the important guiding significance to our research. We have studied comments carefully and have made corrections which we hope meet with approval. Line numbers used in the reply section refer to the text in the revised manuscript.

The main corrections in the paper and the responds to the reviewer’s comments are as following:

Reviewer #2

The paper is interesting and addresses a current issue of great significance in everyday life.
In this study, a set of historical loess landslide samples was first constructed for deep learning. The samples were from three study sites, consisting mainly of old landslides with a long slip time. The open source Google Earth image was used as a data source, and three deep learning methods for detecting objects, namely MASKR-CNN, RetinaNet and YOLO v3, were used to automatically identify old loess landslides.
The results presented by the authors indicated that, based on the deep learning method, Google Earth images can be used to automatically identify most historical loess landslides in the studied sites, but there are also problems of harmonizing images or rather their resolutions. , especially since images of different resolutions are used from different satellites and I recommend that this method be used on a wide scale, but in parallel with other established methods.
The identification of historical landslides is the basis of the regional management of landslide risk, the images of Google Earth being more recent and they must be corroborated with older data to obtain a multicriteria management and have a solid history.
Early identification of landslides can also be done in combination with other monitoring technologies, as many landslides are grouped and reborn and early identification can prevent many disasters that occur due to lack of real-time landslide monitoring.
Especially today and in the age of big data that can be updated and processed, using the deep learning method to identify landslides can substantially increase efficiency, which is of great importance for preventing and mitigating landslides.
I congratulate the authors of the article and it is a research that helps the management progress in monitoring landslides.
The article can be published in the journal.

Response: Thank you very much for your affirmation.

Reviewer 3 Report

remotesensing-1531838-peer-review-v1

The manuscript “Loess landslide detection using object detection algorithms in Northwest China” addresses an interesting and up-to-date subject, which adheres to Remote Sensing journal policies.

The manuscript tackles an interesting topic, and presents a good remote sensing application. In addition, the work is well-conceived and written. The manuscript needs some improvements before resubmission:

  • Formatting based on Mdpi template, especially the citations
  • A figure with the position of the landslides used for training, validation, and testing would be beneficial
  • Discussion chapter needs improvements, as well as citations to similar research and findings

Author Response

Dear Editors and Reviewers:

Thank you for your letter and for the reviewers’ comments concerning our manuscript entitled “Loess landslide detection using object detection algorithms in Northwest China” (ID: remotesensing-1531838). Those comments are all valuable and very helpful for revising and improving our paper, as well as the important guiding significance to our research. We have studied comments carefully and have made corrections which we hope meet with approval. Line numbers used in the reply section refer to the text in the revised manuscript.

The main corrections in the paper and the responds to the reviewer’s comments are as following:

Reviewer #3

The manuscript “Loess landslide detection using object detection algorithms in Northwest China” addresses an interesting and up-to-date subject, which adheres to Remote Sensing journal policies.

The manuscript tackles an interesting topic, and presents a good remote sensing application. In addition, the work is well-conceived and written. The manuscript needs some improvements before resubmission:

Formatting based on Mdpi template, especially the citations

A figure with the position of the landslides used for training, validation, and testing would be beneficial

Discussion chapter needs improvements, as well as citations to similar research and findings

Response: Thank you very much for your constructive suggestion. We have optimized the position figure, added the analysis on the area of detection and omission for the three model, and citated similar researches.
